# A Cross-Sectional Analysis of the Nicotine Metabolite Ratio and Its Association with Sociodemographic and Smoking Characteristics among People with HIV Who Smoke in South Africa

**DOI:** 10.3390/ijerph20065090

**Published:** 2023-03-14

**Authors:** Chukwudi Keke, Zane Wilson, Limakatso Lebina, Katlego Motlhaoleng, David Abrams, Ebrahim Variava, Nikhil Gupte, Raymond Niaura, Neil Martinson, Jonathan E. Golub, Jessica L. Elf

**Affiliations:** 1Department of Environmental and Radiological Health Science, Colorado State University, Fort Collins, CO 80523, USA; 2Africa Health Research Institute, Somkhele, Myeki 3935, South Africa; 3Perinatal HIV Research Unit, Soweto 1862, South Africa; 4School of Global Public Health, New York University, New York, NY 10003, USA; 5Klerksdorp Tshepong Hospital Complex, Matlosana 2574, South Africa; 6Department of Infectious Diseases, Johns Hopkins School of Medicine, Baltimore, MD 21205, USA

**Keywords:** HIV, smoking, nicotine metabolite ratio, NMR, South Africa

## Abstract

The nicotine metabolite ratio (NMR) is associated with race/ethnicity but has not been evaluated among smokers in the African region. We conducted a cross-sectional analysis of baseline data from a large randomized, controlled trial for smoking cessation among people with HIV (PWH) in South Africa. Urine samples were analyzed for the NMR and evaluated as a binary variable using a cutoff value of the fourth quartile to determine the fastest metabolizers. The median NMR was 0.31 (IQR: 0.31, 0.32; range: 0.29, 0.57); the cut-point for fast metabolizers was ≥0.3174 ng/mL. A high NMR was not associated with the number of cigarettes per day (OR = 1.10, 95% CI: 0.71, 1.70, *p* = 0.66) but was associated with 40% lower odds of a quit attempt in the past year (OR = 0.69; 95% CI: 0.44, 1.07, *p* = 0.09) and alcohol use (OR = 0.59, 95% CI: 0.32, 1.06, *p* = 0.07). No association was seen with marijuana or HIV clinical characteristics. As we found only minimal variability in the NMR and minimal associations with intensity of smoking, NMR may be of limited clinical value in this population, although it may inform which individuals are less likely to make a quit attempt.

## 1. Introduction

Despite recent advancements in antiretroviral therapy (ART), HIV remains a significant threat to global health, with 41 million people living with HIV and 1.5 million new infections each year [1]. In general, people with HIV (PWH) have a higher smoking prevalence than the general population [2,3,4,5], but they also have fewer resources for, and are less successful at, quitting. Smoking increases morbidity and mortality among PWH, who are already at high risk for smoking-related illnesses, including malignancies, tuberculosis (TB), heart disease, and other pulmonary diseases [6,7,8,9]. South Africa has the highest global burden of HIV, with an estimated 7.5 million PWH and an HIV prevalence of 19% [1], and 52% of men and 13% of women with HIV in South Africa are current smokers [10]. Given the racial/ethnic makeup of the South African population and the historical inequality perpetuated by apartheid, the burden of HIV in South Africa falls largely on Black South Africans.

The nicotine metabolite ratio (NMR) is a novel biomarker that characterizes how fast the body metabolizes nicotine [11,12,13,14,15,16,17]. NMR is the ratio of 3′-hydroxycotinine (3-HC) to cotinine, both of which are metabolized by the liver enzyme CYP2A6 [11,12,13,14,15,16,17]. A higher NMR represents faster metabolism of nicotine and has been associated with a greater consumption of nicotine and a greater difficulty in quitting smoking [11,12,13,14,15,16,17]. People with a higher NMR also exhibit reduced efficacy of nicotine patches for smoking cessation but a higher efficacy of varenicline; NMR has been evaluated as a tool to inform the assignment of treatment with these pharmacotherapies [11,12,13,14,15,16,17]. Existing research on NMR, which has predominantly been conducted in the United States, indicates that African Americans appear to have lower NMR, compared to those of Caucasian descent, but no studies have been conducted on the African continent, nor in other low- and middle-resource settings [12,13,18]. Describing the NMR in the South African population, especially among PWH, may determine whether this biomarker can be used to inform smoking cessation strategies in this population. This present study aims to evaluate the variability of NMR and its association with sociodemographic, tobacco use, and clinical characteristics among PWH in South Africa.

## 2. Materials and Methods

We evaluated baseline data from a randomized, controlled trial for smoking cessation among PWH in South Africa. Ethics approval for the primary randomized control trial was provided by the University of Witwatersrand Ethics Committee and the Johns Hopkins University Institutional Review Board. Approval for the current analysis of urine samples was approved by the Colorado State University Institutional Review Board. Written informed consent was obtained from all participants in the randomized control trial.

The primary randomized, controlled trial consisted of behavioral counseling versus behavioral counseling plus nicotine replacement therapy [19]. Briefly, a total of 561 participants were recruited from three HIV clinics in Matlosana, South Africa: Grace Mokgomo, Jouberton, and Tshepong HIV Wellness Clinic. Located in the Northwest Province in central South Africa, Matlosana is a peri-urban district, with a population of approximately 425,000 and an HIV antenatal prevalence of 30% [20]. Participants were eligible for the randomized, controlled trial if they were at least 18 years of age, current daily smokers, and had an HIV infection. Participants were excluded from the primary trial if they had a history of adverse interactions with nicotine replacement therapy or if they were currently using smokeless tobacco, electronic cigarettes, or other nicotine replacement therapies. Self-reported smoking was biochemically verified with a point-of-care urine cotinine test and an exhaled breath carbon monoxide (CO).

At baseline, sociodemographic and smoking behaviors, including sex, age, employment status, education, and household income, were collected. Smoking behavior variables included the average number of cigarettes per day, time from last cigarette, time from waking to first cigarette, and secondhand smoke exposure at work and home. Responses were used to calculate the Heaviness of Smoking Index as a measure of nicotine addiction [21]. Alcohol misuse was characterized using the CAGE Substance Abuse Screening Tool [22], and participants self-reported current marijuana use. A urine sample was collected from all consenting participants. Urine samples were analyzed for NMR at the University of Witwatersrand Clinical Laboratory Services in Johannesburg, South Africa by liquid chromatography tandem mass spectrometry (LC/MS/MS) using a U.S. Centers for Disease Control and Prevention Laboratory Protocol [23]. Consistent with methods in literature [24,25,26], and to account for the potential difference in the range of NMR levels in this racial/ethnically homogenous population, NMR levels were evaluated as a dichotomized variable (high versus normal metabolizers), using a cut-off of the fourth quartile, as well as a log-transformed continuous variable.

### Statistical Analysis

Sociodemographic, smoking, and clinical characteristics were described using the median and the interquartile range (IQR) for continuous variables and counts, with percentages for categorical variables. We compared fast to normal nicotine metabolizers using the Wilcoxon rank-sum test for continuous variables and the Chi (ꭓ^2^) squared test of independence for categorical variables. The associations between sociodemographic, smoking, and clinical characteristics and high (versus normal) NMR were evaluated using univariate logistic regression; odds ratios (OR) were reported alongside their corresponding 95% confidence intervals (CI). Associations between NMR as a log-transformed continuous variable and characteristics of interest were evaluated using univariate linear regression models. Stratified analysis was also used to evaluate measures of association separately for males and females. All tests presented were two-sided, with a *p*-value of 0.05 considered statistically significant. The R software (version 4.2.1) was used to conduct all statistical analyses.

## 3. Results

Of the 561 participants in the primary trial, 438 consented to provide urine samples and were included in this analysis. We found no difference in sociodemographic, tobacco use, or clinical characteristics between those included in this analysis and those in the primary trial (Table 1). Most participants were males (n = 343; 76%), under the age of 40 years (n = 190; 57%), had less than a 12th grade education (n = 366; 84%), were unemployed (n = 324; 66%), and had a monthly family income of less than ZAR 1000, or approximately USD 58 (n = 183; 42%). Among those with viral load data, most participants had a viral load >200 copies/mL (n = 54; 75%) and a CD4+ T cell count between 200–500 cells/µL (n= 138; 50%), and most participants did not have TB (n = 419; 96%). At the time of data collection, n = 203 (84%) participants were not on antiretroviral therapy. Most participants also showed moderate addiction to tobacco (n = 330; 76%) and alcohol misuse (n = 236; 79%), as defined by the Heaviness of Smoking Index and the CAGE substance abuse tool, respectively. The median NMR was 0.3174 ng/mL (IQR: O.31, 0.32, range: 0.29–0.57); we classified the bottom 75% of the sample (n = 328) as normal metabolizers (<0.3174 ng/mL) and the top 25% of the sample (n = 110) as fast metabolizers (≥0.3174 ng/mL), given that the variability in measurements was predominantly in the top quartile of values.

In the univariate analysis, NMR was moderately associated with sociodemographic variables; however, the direction of association was inconsistent. Individuals with more than a twelfth-grade education were 76% more likely to be fast metabolizers than those who had less than a twelfth-grade education (OR = 1.76; 95% CI: 1.01, 3.0; *p* = 0.04), but those who had a monthly family income of more than ZAR 1000 were less likely to be fast metabolizers than those who had a monthly family income of less than ZAR 1000 (OR = 0.82, 95% CI: 0.53, 1.27; *p* = 0.38). Compared to those below the age of 40, individuals above the age of 40 were less likely to be fast metabolizers (OR = 0.83; 95% CI: 0.53, 1.28; *p* = 0.39).

A high NMR had limited association with more intense smoking characteristics. Individuals with a quit attempt in the past year were less likely to be fast metabolizers, compared to those who never attempted to quit (OR = 0.69, 95% CI: 0.44, 1.07, *p* = 0.09), and those exposed to secondhand smoke at work were 88% more likely to be fast metabolizers than those not exposed to secondhand smoke at work (OR = 1.88, 95% CI: 0.82, 4.63, *p* = 0.15). We found no association between cigarettes per day and high NMR (OR = 1.10, 95% CI: 0.71, 1.70, *p* = 0.66). Alcohol misuse was negatively associated with NMR; individuals who reported alcohol misuse were less likely to be fast metabolizers than those without alcohol misuse (OR = 0.59, 95% CI: 0.32, 1.06, *p* = 0.07). Only weak to no associations were seen between NMR and marijuana use or HIV clinical characteristics.

Separate linear regression models were used to examine the relationship between NMR as a log-transformed continuous variable and each characteristic of interest; results were consistent with associations found with our binary outcome of fast versus normal metabolizers. No differences were seen in the association between NMR and variables of interest when the analysis was stratified by sex.

## 4. Discussion

This cross-sectional study described the variability of NMR and its association with sociodemographic, tobacco use, and clinical characteristics in a population of PWH who smoke in South Africa. NMR was dichotomized into fast and regular metabolizers using a cut-off of the fourth quartile; few associations were seen between NMR and variables of interest. NMR was minimally associated with tobacco use characteristics, and the NMR had weak to no association with sociodemographic or clinical characteristics. Our study sample, which was reflective of the burden of smoking among PWH in South Africa, was a racially and ethnically homogenous population of Black South Africans, primarily of low socioeconomic status; this likely contributed to the observed relative lack of variability in NMR values.

We reported little variability in values of NMR in this population, and we found a minimal association between fast NMR and more intense smoking characteristics, which was inconsistent with evidence from smokers from both the general population and among PWH [11,18,27]. We found no evidence that those with a fast NMR tended to smoke more cigarettes per day; our observed association was inconsistent with existing literature, which showed a positive association between high NMR and a higher number of cigarettes per day [11,18,27]. Although we also found a negative association between NMR and quit attempts in the last year, there was limited to no association between NMR and nicotine dependence, as measured by the Heaviness of Smoking Index. We also found that those reporting alcohol misuse were less likely to be fast metabolizers of nicotine. This was inconsistent with recent studies in the general population [27,28], which reported positive associations between alcohol consumption and NMR. Similarly, in a trial of men receiving treatment for alcohol use disorder, a 50% reduction in NMR was reported among those who successfully abstained from alcohol, compared to the baseline, despite little change in tobacco use, suggesting that chronic alcohol exposure induces CYP2A6 activity, which is responsible for nicotine metabolism [29].

Overall, the values of NMR found in our study sample were lower than what was reported in other populations of PWH in high-resource settings; however, the existing evidence came largely from more diverse Western settings, with a large proportion of participants of Caucasian descent. Mean values of NMR in literature among PWH range from 0.35 to 0.47 [11,25,27,30], whereas the mean in this study was 0.32. Given our sample was entirely Black South African, and given African Americans on average have a lower NMR than other racial/ethnic groups, the homogeneity of our population, with a genetic makeup closer to those with typically lower NMRs, aligned our findings with these other studies. The variability in the NMR observed in this study may be too minimal to allow the use of NMR as a tool for clinical decision-making at the individual level in this population. Existing studies among PWH reported greater variability in NMR, with values ranging from 0.054 to 1.42 [11], while our minimum and maximum values were 0.29 to 0.57, respectively. In addition, participants of African descent in other studies also reported a higher average number of cigarettes per day than our study population, with an average number of cigarettes per day of 16.6 among individuals of African descent [18], compared to 11.1 in this present study. Existing studies largely found no association between NMR and sociodemographic characteristics other than sex, race/ethnicity, and age; associations in our study between NMR and sex, age, and other sociodemographic characteristics were weak and inconsistent in direction. A handful of studies in the general population demonstrated a relationship between high NMR and older age [24,26,31]. However, studies in HIV populations did not report these associations [11,30], which was consistent with our findings. We also found no association with sex, which was inconsistent with existing evidence showing that women have a higher NMR than men [18,27].

Our findings are not without limitations. NMR was measured in urine in this study, which tends to correlate imperfectly with NMR measured in plasma [32,33]. NMR values obtained in our study may not accurately represent the sample or the population but is likely to be a reasonable approximation of plasma NMR. Recent evidence also suggests that some antiretroviral medications may impact nicotine metabolism. Efavirenz was found to raise NMR, given it is metabolized by the same pathway as nicotine [13]; this implies that people with HIV who smoke and are on efavirenz may have more intense smoking behaviors and more difficulty quitting smoking compared to those on other ARTs who are given an elevated NMR. However, our sample was drawn from a population with limited access to antiretroviral drugs at the time of recruitment, given policies at the time limiting the initiation of ART to individuals with a CD4+ count under a defined level, compared to current universal test and treat policies. Additionally, other predictors of smoking behavior, such as extreme poverty, may have influenced smoking behavior in our sample to a degree that would have reduced the impact of NMR on smoking behavior [26]. Given that included subjects were recruited for participation in a clinical trial for smoking cessation, our sample may have included a disproportionate number of individuals with a slower metabolism. Participants in the primary trial must have been willing to set a quit date, and those with faster metabolisms may have been less willing to attempt to quit and therefore less willing to participate in the primary cessation study; this may have limited the generalizability of our findings. While a more racially/ethnically diverse sample may result in the greater variability of NMR, our study sample is reflective of the burden of HIV in this setting. The results of this study are most generalizable to ethnically African populations with HIV in the region; however, evaluations of those without HIV in this population would be informative.

## 5. Conclusions

We found minimal variability in NMR values among a population of Black South Africans with HIV, compared to typical ranges found in other more diverse settings. While there was some association between NMR and smoking characteristics in this population, the clinical utility may be the strongest for understanding the willingness to make a quit attempt among individuals with higher NMR. Overall, our findings highlighted the potential limitations of using NMR to inform approaches for treatment of nicotine dependence among PWH in this population. Further research is needed to fully elucidate the relationship between NMR and a successful quit attempt to more fully understand the clinical implications of NMR in this setting.

## Figures and Tables

**Table 1 ijerph-20-05090-t001:** Sociodemographic, tobacco use, and clinical characteristics and their univariate association with NMR among adults who smoke and live with HIV in South Africa.

	Total (n = 561)	Total with Urine (n = 438)	Normal NMR (<0.3174) (n = 328)	High NMR (≥0.3174) (n = 110)	*p*-Value	Univariate OR (95% CI)	*p*-Value
Sociodemographic							
Sex							
Female	123 (22)	95 (22)	71 (22)	24 (22)	1.00	REF	
Male	438 (78)	343 (78)	257 (78)	86 (78)		0.98 (0.59, 1.69)	0.96
Age, median (IQR)	38 (31, 46)	38 (31, 45)	38 (31, 45)	37 (30, 46)	0.63	0.99 (0.97, 1.01)	0.49
Age							
<40	317 (57)	247 (57)	181 (55)	66 (60)	0.60	REF	
≥40	243 (43)	190 (43)	146 (45)	44 (40)		0.83 (0.53, 1.28)	0.39
Education							
Below 12th grade	472 (84)	366 (84)	281 (86)	85 (77)	0.06	REF	
12th grade or above	89 (16)	72 (16)	47 (14)	25 (23)		1.76 (1.01, 3.0)	0.04
Employment							
Unemployed	416 (74)	324 (74)	243 (74)	81 (74)	1.00	REF	
Employed	145 (26)	114 (26)	85 (26)	29 (26)		1.02 (0.62, 1.66)	0.93
Total monthly family income							
≤1000 R	234 (42)	183 (42)	133 (41)	50 (45)	0.44	REF	
>1000 R	326 (58)	254 (58)	194 (59)	60 (55)		0.82 (0.53, 1.27)	0.38
Tobacco Use Behavior							
Heaviness of Smoking Index							
Low addiction	144 (26)	107 (24)	82 (25)	25 (23)	0.71	REF	
Moderate addiction	416 (74)	330 (76)	245 (75)	85 (77)		1.14 (0.69, 1.92)	0.62
Cigarettes per day							
<10 cigarettes	277 (49)	207 (47)	157 (48)	50 (45)	0.74	REF	
≥10 cigarettes	284 (51)	231 (53)	171 (52)	60 (55)		1.10 (0.71, 1.70)	0.66
Motivation to quit smoking, median (IQR)	10 (9, 10)	10 (8, 10)	10 (8, 10)	10 (8, 10)	0.84	1.05 (0.92, 1.21)	0.51
Quit attempt in the past year							
No	213 (38)	158 (36)	111 (34)	47 (43)	0.12	REF	
Yes	348 (62)	280 (64)	217 (66)	63 (57)		0.69 (0.44, 1.07)	0.09
Exposed to secondhand smoke at home							
No	378 (68)	295 (68)	221 (68)	74 (68)	1.00	REF	
Yes	181 (32)	141 (33)	106 (32)	35 (32)		0.99 (0.62, 1.56)	0.95
Exposed to SHS at work ^a^							
No	67 (12)	51 (37)	42 (40)	9 (26)	0.21	REF	
Yes	113 (20)	87 (63)	62 (60)	25 (74)		1.88 (0.82, 4.63)	0.15
Exhaled Breath CO (ppm)	15 (9, 22)	15 (9, 22)	15 (9, 22)	17 (9, 23)	0.27	1.00 (0.98, 1.02)	0.79
Smokescreen baseline analysis							
Light	356 (70)	274 (70)	207 (70)	67 (69)	0.98	REF	
Moderate	89 (17)	67 (17)	50 (17)	17 (18)		0.99 (0.98, 1.02)	0.96
Heavy	65 (13)	53 (13)	40 (13)	13 (13)		1.01 (0.99, 1.03)	0.20
Other Substance Use							
Alcohol consumption							
No alcohol misuse	91 (23)	63 (21)	41 (19)	22 (28)	0.10	REF	
Alcohol misuse	302 (77)	236 (79)	180 (81)	56 (72)		0.59 (0.32, 1.06)	0.07
Marijuana use							
No	179 (60)	138 (59)	100 (58)	38 (63)	0.55	REF	
Yes	120 (41)	95 (41)	73 (42)	22 (37)		0.79 (0.43, 1.44)	0.45
Clinical Characteristics							
Current Viral Load							
<200 copies/mL	22 (25)	18 (25)	15 (25)	3 (25)	1.00	REF	
>200 copies/mL	65 (75)	54 (75)	45 (75)	9 (75)		1.0 (0.25, 4.94)	1.00
Current CD4+ T cell count							
<200 cells/µL	85 (25)	69 (25)	52 (25)	17 (28)	0.88	REF	
200–500 cells/µL	159 (48)	138 (50)	103 (49)	29 (48)		0.86 (0.44, 1.73)	0.67
>500 cells/µL	88 (27)	70 (25)	55 (26)	15 (24)		0.83 (0.37, 1.84)	0.53
High Blood Pressure Medication							
No	499 (89)	391 (89)	292 (89)	99 (90)	0.91	REF	
Yes	62 (11)	47 (11)	36 (11)	11 (10)		0.90 (0.42, 1.78)	0.77
Current TB							
No	539 (96)	419 (96)	313 (96)	106 (96)	1.00	REF	
Yes	21 (4)	18 (4)	14 (4)	4 (4)		0.84 (0.23, 2.41)	0.77
Current Cough							
No	338 (60)	270 (62)	200 (61)	70 (64)	0.70	REF	
Yes	223 (40)	168 (38)	128 (39)	40 (36)		0.89 (0.56, 1.39)	0.62

^a^ Among those who work (n = 180).

## Data Availability

The data underlying this article will be shared upon reasonable request to the corresponding author.

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
