# Peer review of "A Cross-Sectional Analysis of the Nicotine Metabolite Ratio and Its Association with Sociodemographic and Smoking Characteristics among People with HIV Who Smoke in South Africa"

_ijerph, 2023, doi:10.3390/ijerph20065090_

Round 1
Reviewer 1 Report
Thank you for the opportunity to review the manuscript namely „A cross-sectional analysis of the nicotine metabolite ratio and its association with sociodemographic and smoking characteristics among people with HIV who smoke in South Africa“. This cross-sectional study described the variability of NMR and its association with sociodemographic, tobacco use, and clinical characteristics in a population of PWH who smoke in South Africa. I congratulate the Authors of this manuscript for the good work done.
However, I have some concerns:
Major concerns
Introduction
I propose that the description of this section may be completed for the aim of the study and on the research questions (RQ) of this study.
Materials and Methods
Line 94: The term „Chi (c2)“ must be specified.
Results
Well-written.
Discussion
Line 155: What does it mean: “Moderate associations were seen with...“?
I would suggest that the main findings not only be compared with those of other authors, but also could be interpreted.
Conclusions
This study was quantitative, but it is hardly possible to generalize results based on sample size of this study. Unless the Authors can provide calculations for a representative sample size in the methodology section. Otherwise, this is the limitation of this cross-sectional study.
The conclusions are too abstract. I recommend that the conclusions may be adjusted so that they answer the aim or the research questions (RQ) raised by this study.
In the alternative, I propose to refer to the specific practical recommendations of this study.
Minor concerns:
References must be customized according to IJERPH formatting guidelines.
Please recheck the use of hyphens, en dashes, and em dashes throughout the text.
English must be checked before the paper will be published.
Kind Regards
Author Response
We thank the reviewers for their thoughtful critique of our manuscript. We have carefully considered each comment and hope to have satisfactorily addressed any concerns the reviewers may have had.
Introduction
I propose that the description of this section may be completed for the aim of the study and on the research questions (RQ) of this study.
Author response: The introduction section has been updated, such that it shows the main aim of this study and can be found on lines 59-61 in the revised manuscript.
Materials and Methods
Line 94: The term „Chi (c )“ must be specified
Author response: Thank you for pointing this out. The term “Chi (ê“2) squared” has been corrected on line 106 in the revised manuscript.
Results
Well written
Author response: Thank you!
Discussion
Line 155: What does it mean: “Moderate associations were seen with...“?
Author response: We intend to convey that the observed association is not strong, however, it is certainly noticeable. We have made some slight edits in Line 180 in order to clarify.
I would suggest that the main findings not only be compared with those of other authors, but also could be interpreted.
Author response: Thank you for your suggestion. However, we believe that the main findings are well interpreted in the discussion section. We have included interpretation in the first paragraph of the discussion, throughout the discussion, Lines 199-201, the limitations section, and the conclusion section. We have also made some additional edits in the discussion section which hopefully helps clarify these points.
Conclusions
This study was quantitative, but it is hardly possible to generalize results based on sample size of this study. Unless the Authors can provide calculations for a representative sample size in the methodology section. Otherwise, this is the limitation of this cross-sectional study.
Author response: Thanks for pointing this out. We agree that this is a potential limitation of our study. We have added this as a limitation which can be found on line 236 in the revised manuscript. Further, we have made clarifications about the generalizability of our study in Lines 237-241.
The conclusions are too abstract. I recommend that the conclusions may be adjusted so that they answer the aim or the research questions (RQ) raised by this study. In the alternative, I propose to refer to the specific practical recommendations of this study.
Author response: Thanks for the suggestion. We have updated our concluding statement such that it highlights the practical recommendation of the study, which can be found in lines 247-251 in the revised manuscript.
In addition to the above comments, all spelling and grammatical errors pointed out by the reviewers have been corrected in the revised manuscript.
Reviewer 2 Report
This is an exciting manuscript about the variability of nicotine metabolite ratio (NMR) and its association with sociodemographic and clinical characteristics in people living with HIV/AIDS (PLWH). While the findings are limited, with minimal variability, considering that smoking increases morbidity and mortality among PLWH, the results present relevant clinical applications.
What is the rationale for describing the characteristics of the total (n=561) versus the urine individuals (n=428)? The analysis must be restricted to the 428 individuals from whom a urine sample was obtained.
In this sense, in table 1, the columns Total and Total with urine should be deleted. The p values must be properly described at the top of the table.
Author Response
Comments by review 2
We thank the reviewers for their thoughtful critique of our manuscript. We have carefully considered each comment and hope to have satisfactorily addressed any concerns the reviewers may have had.
What is the rationale for describing the characteristics of the total (n=561) versus the urine individuals (n=428)? The analysis must be restricted to the 428 individuals from whom a urine sample was obtained.
Author response: Describing the characteristics of the total (n=561) versus the urine individuals (n=428) is to give readers more information on the distinction between the participants in the primary trial and those we include in our analysis. Participants must have consented specifically to provide this urine sample, and we wanted to demonstrate whether or not there were any differences between the full baseline sample and the subsample included in this present analysis. We indeed agree with you that the analysis should be restricted to the individuals who provided a urine sample; our analysis was restricted to the 438 individuals that consented to provide urine samples as outlined in the result section in Lines 117-118.
In this sense, in table 1, the columns Total and Total with urine should be deleted. The p values must be properly described at the top of the table.
Author response: While we appreciate the reviewer’s feedback, we respectfully keeping the first column with the total participants in Table 1. We think that having both total and total urine columns in table 1 will help the reader understand the characteristics of the individuals to whom the study findings apply and that there was no meaningful difference between the full sample and the sample included in this present analysis.
The p values must be properly described at the top of the table.
Author response: Thanks for pointing this out. We have properly described the “p-value” in table 1 in the revised manuscript.
Round 2
Reviewer 1 Report
Thank you for the opportunity to re-review the topic „A cross-sectional analysis of the nicotine metabolite ratio and its association with sociodemographic and smoking characteristics among people with HIV who smoke in South Africa“.
I still have a little observation:
In the manuscript title, I suggest that the Authors could change the term “a cross-sectional analysis” to “a cross-sectional study”.
Additionally, practical recommendations could be more targeted and precise.
In general, the Authors have answered my questions. I recommend that the paper can be accepted.
Best Regards